# The test-retest reliability and agreement between a fixed frame and belt-stabilised handheld dynamometer for isometric hip flexion and extension peak force measurement in recreational cyclists

**Dion D'Mello[1], Benn Digweed[1,2], Tom Hughes** [1,3]*

**1** Department of Health Professions, Manchester Metropolitan University, Manchester, United Kingdom, **2** United Kingdom Sports Institute, UK Sports Institute High Performance Centre, Manchester Institute of Health and Performance, Manchester, United Kingdom, **3** Institute of Sport, Manchester Metropolitan University, Manchester, United Kingdom

* t.hughes@mmu.ac.uk

## Abstract

### Introduction

Cycling performance is influenced by hip flexor and extensor muscle strength. While belt stabilised handheld dynamometers (B-HHD) are valid for measuring isometric hip muscle strength, fixed frame dynamometers are becoming popular, offering potentially better stability and reliability. However, the reliability of both devices has not been examined in cyclists. This study evaluated the test-retest reliability and agreement between a B-HHD (MicroFET2, Hoggan Scientific) and a fixed-frame dynamometer (ForceFrame (FF) Max, Vald Performance) for hip flexion and extension peak force measurement in cyclists.

### Methods

A test-retest design was used. Twenty-five recreational cyclists (age±SD: 36.64 (±12.34) years; 22 males) were tested twice, approximately 72 hours apart. Three unilateral maximal voluntary isometric contractions (MVIC) of the hip flexors and extensors of each limb were performed, using the B-HHD and FF in a random order. Within and between session reliability was determined using intraclass correlation coefficients (ICCs) $_{3.1 \& 3.k.}$ Standard error of measurements (SEM) and minimal detectable changes (MDC) were calculated. Agreement was assessed using 95% limits of agreement (LOA).

### Results

For hip flexion, within and between session reliability was good to excellent, and SEMs were similar (B-HHD ICCs = 0.77–0.93, SEMs = 14.25–22.71N (7.19–10.38%);

**Data availability statement:** An anonymised version of the dataset and Stata code is accessible via Manchester Metropolitan University's data repository (E-space), and is freely available at https://doi.org/10.23634/MMU.00643463.

**Funding:** Neither the authors nor Manchester Metropolitan University received funding for this work. No funding or incentives were provided by Vald Performance to conduct this work and the ForceFrame Max used in this study was loaned to Manchester Metropolitan University at no cost.

**Competing interests:** DDM was a student at Manchester Metropolitan University and this work was submitted as a dissertation project for the qualification of MSc Physiotherapy. He was employed at Manchester Movement Unit and British Cycling as a Sports Massage therapist during data collection. The project was conceived during his employment with British Cycling. BD was employed by Manchester Metropolitan University and the UK Sports Institute. TH is employed at Manchester Metropolitan University. This does not alter our adherence to PLOS ONE policies on sharing data and materials.

FF ICCs = 0.77–0.95, SEMs = 7.80N–18.98N (3.47%−8.54%)). FF MDCs were lower within-session (21.61–39.48N (9.60–17.97%)) than B-HHD MDCs (39.50–62.95N (19.94–28.78%)), but similar between-sessions (FF MDCs = 41.25–52.61N (19.42–23.66%); B-HHD MDCs = 41.21N–48.95N (18.53–23.77%)).

For hip extension, both devices demonstrated good to excellent reliability and SEMs were similar (B-HHD ICCs = 0.90–0.95, SEMs = 15.77–21.53N (7.38–9.96%); FF ICCs = 0.85–0.95, SEMs = 19.21–29.05N (7.82–11.78%) within and between sessions). All LOA exceeded a 20N acceptability threshold.

## Conclusion

Both devices are reliable in recreational cyclists, but large MDCs suggest that caution is needed when interpreting repeated measurements. Due to poor agreement, the devices are not interchangeable so should be considered device specific. In practice, our preliminary results suggest FF data cannot be compared with B-HHD data and vice versa, so the same device should be used for repeated measurements in this population.

## Introduction

Cycling places physiological stress on the metabolic, cardiovascular and musculoskeletal systems, which can improve cardiovascular fitness [1], cognitive function and overall wellbeing [2]. It is associated with a reduced risk of cardiovascular disease (CVD) and related risk factors (including obesity, hypertension and elevated triglycerides) [3]. Cycling is linked to a 17% reduction in CVD mortality [3] and a 20% reduction in all-cause mortality [1], making it a highly recommended activity for public health improvement [4]. Additionally, cycling has become increasingly popular in the United States (US) [5,6] and other European countries [7,8] for transportation, recreational [5] and competitive purposes [6].

Approximately 85% of cycling power is generated during the pedal downstroke [9] by the hip extensor, knee extensor, and ankle plantarflexor muscles [10]. During pedal upstroke, power is generated by the hip and knee flexors, with biarticular muscles maintaining joint stability [11]. To enhance muscular strength and cycling performance, recreational and competitive cyclists commonly combine resistance training with cycling endurance training [12]. Indeed, resistance training has been shown to improve peak power output in competitive cyclists [13,14], cycling economy and VO$_2$ Max in moderately trained cyclists [15,16] and pedalling cadence efficiency in recreational athletes [14,16,17]. However, these latter benefits are less clear in highly trained cyclists [12,14–16].

Given the importance of hip strength in cycling performance and the risk of injury associated with cycling participation [18], regular muscle strength assessments may provide valuable data for cyclists, coaches and health professionals. Indeed, regular physical testing can be useful for performance tracking, training evaluation and planning [19–21], setting rehabilitation targets [19] and assessing treatment effectiveness following injury [22].

Isokinetic dynamometers (IKD) are considered the gold standard for muscle strength assessment due to their reliability and ability to measure various strength components (including peak force, work, power, endurance and force curves) [22]. However, IKDs are expensive, non-portable and require expert training, limiting their use to clinical [23,24] and research settings. In contrast, portable and simple hand-held dynamometers (HHDs) (such as the MicroFET2, Hoggan Scientific, Salt Lake City, USA) are commonly used to measure isometric strength, due to their cost-effectiveness, mass-testing capabilities [25], accessibility, and ease of use without specialist training [26]. HHDs can quantify maximal isometric voluntary contractions (MVICs), which are valid assessments of maximal strength [21].

HHDs show acceptable intra and inter-rater reliability for measuring isometric hip flexion and extension MVICs (weighted correlation coefficient (WCC) ranges: 0.79–0.91 and 0.81–0.87, respectively) [27]. However, measurements can be influenced by the examiner's strength, potentially causing inter-rater bias [28], and the lack of device stabilisation can introduce errors [29]. External fixation methods, such as belt stabilisation, can eliminate examiner-provided resistance and improve test-retest reliability (WCC ranges = 0.91–0.93, $ICC_{2.1}$ = 0.91–0.95) [27,30]. Importantly, population characteristics influence device reliability [31,32], but to date, the reliability of external, belt-stabilised HHDs (B-HHDs) has not been investigated in cyclists, so it is unclear if these estimates apply to this population.

While belt stabilisation may improve reliability, newer fixed frame dynamometers (such as the Vald ForceFrame Max (Vald Performance, Brisbane, Australia)), may offer superior stabilisation because sensors are mounted directly to a frame and base. Fixed frame devices are also highly adjustable, which allows testing of most muscle groups. A different iteration of the ForceFrame (i.e., ForceFrame Fold) has shown good to excellent test-retest reliability for hip adduction ($ICC_{2.1}$ = 0.81–0.90) and moderate to excellent test-retest reliability for hip abduction MVICs ($ICC_{2.1}$ = 0.72–0.92) in field sport athletes [33]. However, to the best of our knowledge, no previous studies have investigated the reliability of any ForceFrame version for measuring hip flexion and extension MVICs in any population.

Given the growing preference to replace HHDs with fixed frame devices, [34] it is essential to determine whether measurements from both devices can be interchanged.a Establishing this would help ensure consistent and accurate data collection in settings where both devices are used, where longitudinal B-HHD data already exist and a fixed-frame device is being introduced, or when practitioners are deciding which measurement method to use.

The aim of this study was to evaluate the test-retest reliability and agreement between a belt-stabilised MicroFET2 HHD and the ForceFrame Max, for measurement of hip flexion and extension MVIC peak force in recreational cyclists.

## Materials and methods

This study was reported according to The Guidelines for Reporting of Reliability and Agreement Studies [32] and the Consensus-based Standards for the Selection of health Measurement INstruments (COSMIN) Reporting Guideline for Studies on Measurement Properties of Patient Reported Outcomes Measures [35]. The methodology has been detailed *a priori*, in a protocol available on Open Science Framework [36].

### Study design and setting

Because both devices were stabilised, examiner effects were considered negligible; therefore, a test–retest design was employed. Data were collected between 04/03/2024 and 22/03/2024 in a laboratory at Manchester Metropolitan University. The examiner was a Physiotherapy MSc student [36], with 3 years of experience using HHDs but no prior experience with the ForceFrame Max. Ten hours of pilot testing was completed for protocol refinement and familiarisation.

### Participants

Recreational and competitive cyclists were recruited using purposive sampling. Posters were placed in cycle shelters at Manchester Metropolitan University and distributed to three cycling clubs in the Greater Manchester area. Potential participants voluntarily contacted the lead researcher for eligibility confirmation and study details.

Participants were included if they were aged >18 years and completed > 150 minutes cycling per week at 'vigorous' intensity (> 10 miles per hour as defined by the Centre for Disease Control and Prevention [37]).

Participants were excluded if they: had an active neurological, cardiovascular or respiratory disease; were pregnant; had any lacerations, abrasions, or contusions at the distal femur affecting the area of HHD and FF load cell sensor application during testing; reported any musculoskeletal injury or orthopaedic surgery to the lumbar spine, pelvis or lower limbs or suffered any systemic illness within 3 months of the first assessment day.

Ethical approval was granted by the Faculty of Health and Education Research Ethics and Governance Committee at Manchester Metropolitan University (no.59674) and the study was conducted in accordance with the 2013 Declaration of Helsinki. All participants provided written informed consent prior to participation.

## Sample size

Sample size was calculated using the confidence interval (CI) width procedure of the ICC & SEM Power Sample Size Decision Assistant (https://iriseekhout.shinyapps.io/ICCpower) [38]. Previously reported intraclass correlation coefficents (ICCs) for hip flexion and extension strength measurements were 0.91 and 0.90 for HHDs with external stabilisation [27] and 0.77–0.86 for the Groinbar (Vald Performance, Brisbane, Australia) [23], a similar fixed frame device to the Forceframe Max. For a two-way mixed effects ICC model that is appropriate for a test-retest design [39] as used in this study, our calculation assumed a conservative expected correlation of 0.80, a moderate variance of 10 for between-participant test scores with no expected systematic differences, and aimed for a target CI width of 0.30. Our calculation indicated that recruitment of 25 participants would enable ICC estimation with CI precision of 0.26 for within-session analyses (three trials per testing session), and 0.33 for between-session analyses (mean of three trials compared over two testing sessions) (see S1 Appendix for full calculation).

## Instrumentation and measurement parameters

For all participants, self-reported weekly cycling activity level (minutes) and the usual mode of cycling participation were collected. Baseline measurements of standing height (centimetres) and body mass (kilograms) were recorded on the first testing day using the Leicester Height Measure (Seca Ltd., Birmingham, UK) and the Personal Floor Scale MPE 250K100HM (Kern & Sohn, Germany).

A belt-stabilised Micro-FET2 HHD (B-HHD) and ForceFrame Max Strength Testing System (FF) were used to assess hip flexion and hip extension peak force during MVICs, recorded directly into a laptop computer. FF measurements were transferred to an offline version of the Vald Uploader Manager (Vald Performance, Brisbane, Australia) using a wired connection. B-HHD measurements were manually entered into an Excel (Microsoft, Washington, USA) spreadsheet by the examiner. The B-HHD was calibrated at the start of each testing day. The FF was zeroed between each test. Peak force from all MVICs were recorded in newtons (N) and retained on their continuous scale for analyses.

## Experimental procedure

All data were collected by the primary researcher, who was unblinded. Before each testing session, participants completed a standardised 5-minute warm-up on a stationary bicycle at a self-determined moderate intensity. Across both testing days, participants completed unliateral hip strength assessments with the B-HHD and FF, for both limbs. To eliminate order effects, tests were randomised by muscle group, devices and limb, using the RAND function in Excel, conducted by an independent person.

For B-HHD hip flexion, participants were seated upright on an examination table with their arms crossed against their chest. For FF hip flexion, due to device constraints, testing was conducted with participants in supine with arms crossed against their chest. The examiner used a goniometer to ensure the hip was positioned to 90° flexion for both devices. The sensor for each device was positioned along the distal femur, 1 inch superior to the patellar base.

For B-HHD and FF extension, participants lay prone with their hands under their forehead. For both devices, the examiner used a goniometer to ensure the knee of the tested limb was flexed to 90°. The sensor of each device was positioned along the distal femur, 1 inch superior to the popliteal crease. All testing positions are presented in S2 Appendix. The individuals pictured in S2 Appendix have provided written informed consent (as outlined in PLOS consent form) to publish their image.

For the FF tests, the crossbar height was reproduced on the second testing day. For all B-HHD tests, the fixation belt was secured to the dynamometer and examination table. The examiner manually stabilised the B-HHD to eliminate slipping but did not exert any downward pressure on the sensor.

For each muscle group, device and limb, participants completed three practice trials at 50%, 70% and 90% of their self-perceived maximal efforts. Following these practice attempts, participants performed three 5-second MVICs. For each trial, the examiner provided a standardised instruction of "Begin now – 3, 2, 1 - GO - 1, 2, 3, 4, 5 - relax". A 10-second rest period was given between trials and 2-minute rest periods were provided upon test completion to minimise fatigue and allow for repositioning.

Trials were considered invalid and repeated if: participants altered their test position; the B-HHD slipped; an error was detected on the VALD interface. Participants were withdrawn if they experienced pain in their lower limbs or lumbar spine during testing. Upon completion, data collection was repeated 72 hours later, or as close as possible to 72 hours if participants could not attend at that time.

## Statistical analyses

Descriptive statistics (mean ± standard deviation (SD)) were calculated for height, body mass, cycling volume and age. Proportions were calculated for biological sex and cycling modes. The distribution of raw data was visually inspected using quantile quantile (QQ) plots.

For within-session analyses, peak force (N) was compared across all three trials for each device, muscle group and limb on each testing day. For between-session analyses, the mean of peak force measurements was calculated using all three trials for each participant according to device, muscle group, limb and testing day. Mean values were then compared across testing days.

Two-way mixed effects ICC models are required for test-retest designs [39], so within session $ICCs_{3.1}$ and between session $ICCs_{3.k}$ were calculated with corresponding 95% CIs. ICCs were interpreted as poor < 0.40; fair = 0.40–0.70; good = 0.70–0.90; excellent > 0.90 [40]. Standard error of measurements (SEM) were calculated using the formula $SEM = SD\sqrt{1 - ICC}$. After ICC model fitting, all residuals were also inspected for normality using histograms. Minimal detectable changes (MDC) were calculated using the formula: $MDC = SEM \times 1.96 \times \sqrt{2}$ [41]. SEM and MDC values were reported in newtons (N) and as percentages of mean values.

To determine within-session agreement, the distributions of between device differences were assessed using QQ plots according to muscle group tested, limb and testing day. Bland-Altman plots with 95% limits of agreement (LOA) were constructed [42]. Previously, the MicroFET2 HHD has been shown to have concurrent validity with a similar fixed frame dynamometer system (KangaTech KT360, KangaTech, Melbourne, Australia), with a reported absolute mean difference of 7.74N (95%CI = 1.86–16.40N) [43]. Therefore, agreement was considered clinically acceptable if the 95%LOA were within 20N. All statistical analyses were initially conducted by DD using SPSS 29 (IBM, New York, USA), and repeated by TH using STATA 18 (StataCorp LLC, Texas, USA) for accuracy.

## Results

### Study Participation and missing data

Of twenty-six participants recruited, one withdrew due to pain during testing and was excluded from the analysis. In the remaining dataset, there were no instances of missing data. The remaining participants included 22 males and 3 females

with the following characteristics (mean ± SD): age 36.64 (±12.34) years; height 180.36 (±7.76) cm; mass 80.25 (± 9.86) kg; cycling activity 294 (± 183.24) minutes per week. Ten participants (40%) primarily cycled for travel, 10 (40%) were recreational road cyclists, 3 (12%) were mountain bikers and 2 (8%) used a stationary cycle at the gym.

## Within-session Reliability

For the within-session analyses, descriptive statistics are presented in Table 1. Peak force data and model residuals were approximately normally distributed across devices, limbs and testing days (S3 Appendix). ICCs, 95% CIs, SEMs and MDCs for each limb, device and testing day are presented in Table 2.

For hip flexion, both devices generally showed comparable good to excellent reliability and similar SEMs. Across both days and limbs, FF $ICCs_{3.1}$ ranged between 0.89–0.95. B-HHD reliability of the left limb on day 2 was slightly lower ($ICC_{3.1}$ = 0.77), but all other values were similar to the FF ($ICC_{3.1}$ = 0.84–0.91). FF SEMs were between 7.80–14.24N (or 3.47–6.48%). Similarly, B-HHD SEMs were between 14.25–22.71N (or 7.19–10.38%). However, FF MDCs were consistently lower (21.61–39.48N (9.60–17.97%)) than B-HHD MDCs (39.50–62.95N (19.94–28.78%)).

For hip extension, both devices showed good to excellent reliability with similar SEM and MDC estimates. Across both days and limbs, FF $ICCs_{3.1}$ were 0.87–0.95 and B-HHD $ICCs_{3.1}$ were 0.90–0.95. FF SEMs were between 18.71–24.22N (7.67–10.02%), and B-HHD SEMs were 15.77–20.98N (7.38–9.76%). FF MDCs were between 51.88–67.12N (21.27–27.77%) and B-HHD MDCs were 43.71–58.16N (20.46–27.07%).

**Table 1. Within-session descriptive statistics for peak force measurements, per device and testing day.**

| Within- session Descriptive Statistics | | Session 1 | | | | Session 2 | | | |
|---|---|---|---|---|---|---|---|---|---|
| | | Left limb | | Right limb | | Left Limb | | Right limb | |
| Measure | System | Mean | SD | Mean | SD | Mean | SD | Mean | SD |
| Hip Flexion | B-HHD | 198.12 | 47.13 | 218.75 | 61.96 | 213.81 | 36.24 | 225.96 | 53.27 |
| | FF | 206.33 | 39.09 | 219.67 | 45.92 | 218.47 | 35.10 | 225.08 | 35.17 |
| Hip Extension | B-HHD | 206.19 | 69.35 | 214.86 | 66.79 | 213.67 | 70.67 | 217.28 | 73.03 |
| | FF | 251.57 | 85.00 | 243.87 | 70.70 | 241.71 | 66.23 | 247.56 | 76.15 |

Key: B-HHD = belt stabilised handheld dynamometer; FF = ForceFrame; SD = standard deviation. Please note that all measurements are in newtons (N).

**Table 2. Within-session $ICCs_{3.1}$, 95%CIs, SEM and MDC estimates for both devices and limbs, according to testing day.**

| Within- session Reliability | | Session 1 | | | | | | | | Session 2 | | | | | | | |
|---|---|---|---|---|---|---|---|---|---|---|---|---|---|---|---|---|---|
| | | Left limb | | | | Right limb | | | | Left Limb | | | | Right limb | | | |
| Muscle group | System | ICC | 95% CI | SEM (%) | MDC (%) | ICC | 95% CI | SEM (%) | MDC (%) | ICC | 95% CI | SEM (%) | MDC (%) | ICC | 95% CI | SEM (%) | MDC (%) |
| Hip Flexion | B-HHD | 0.91 | 0.83–0.96 | 14.25 (7.19) | 39.50 (19.94) | 0.87 | 0.76–0.93 | 22.71 (10.38) | 62.95 (28.78) | 0.77 | 0.60–0.88 | 17.55 (8.21) | 48.65 (22.75) | 0.84 | 0.71–0.92 | 21.51 (9.52) | 59.63 (26.39) |
| | FF | 0.94 | 0.88- 0.97 | 9.77 (4.74) | 27.07 (13.12) | 0.90 | 0.82–0.95 | 14.24 (6.48) | 39.48 (17.97) | 0.89 | 0.79–0.95 | 11.75 (5.38) | 32.58 (14.91) | 0.95 | 0.91–0.98 | 7.80 (3.47) | 21.61 (9.60) |
| Hip Extension | B-HHD | 0.93 | 0.87–0.97 | 18.66 (9.05) | 51.73 (25.09) | 0.90 | 0.82–0.95 | 20.98 (9.76) | 58.16 (27.07) | 0.95 | 0.91–0.98 | 15.77 (7.38) | 43.71 (20.46) | 0.95 | 0.90–0.97 | 16.86 (7.78) | 46.74 (21.51) |
| | FF | 0.95 | 0.90–0.98 | 19.35 (7.69) | 53.63 (21.32) | 0.93 | 0.87–0.97 | 18.71 (7.67) | 51.88 (21.27) | 0.87 | 0.76–0.93 | 24.22 (10.02) | 67.12 (27.77) | 0.93 | 0.87–0.97 | 19.86 (8.02) | 55.05 (22.23) |

Note: SEM and MDC measures are presented in newtons (N), whilst (%) values present the estimates as a proportion relative to mean values. Key: B-HHD = belt stabilised handheld dynamometer; CI = confidence interval; FF = ForceFrame; ICC – intraclass correlation coefficient; MDC = minimal detectable change; SEM = standard error of measurement.

## Between-session reliability

Descriptive statistics, ICCs$_{3.k}$, 95% CIs, SEMs and MDCs are presented in Table 3.

For hip flexion, FF reliability was consistently good (ICC$_{3.k.}$ range = 0.77–0.84). B-HHD reliability was good on the left limb (ICC$_{3.k.}$ = 0.81) and excellent on the right (ICC$_{3.k.}$ = 0.93). For both devices, SEMs and MDCs were similar (SEMs: FF = 14.88–18.98N (7.01–8.54%), B-HHD = 14.87–17.66N (6.69–8.57%); MDCs: FF = 41.25–52.61N (19.42–23.66%), B-HHD 41.21N–48.95N (18.53–23.77%)).

For hip extension, both devices had good to excellent reliability across limbs (FF ICC$_{3.k}$ = 0.85–0.93, B-HHD ICC$_{3.k.}$ = 0.90–0.92). SEMs and MDCs were also comparable (SEM: FF = 19.21–29.05N (7.82–11.78%), B-HHD = 19.48–21.53N (9.28–9.96%); MDCs: FF = 53.26–80.51N (21.68–32.64%), B-HHD 54.01–59.67N (25.73–27.62%)).

## Between- device agreement

The distribution of between-device differences was approximately normal across limbs, testing days and muscle groups (S4 Appendix). Bland and Altman plots are presented for hip flexion (Fig 1) and extension (Fig 2). All estimates for mean differences and LOA are presented in S5 Appendix.

For hip flexion, mean agreement ranged from −0.88 to 8.22 N across limbs and testing days. For hip extension, the FF consistently produced higher peak force measures (mean agreement range = 28.04 to 45.38 N) across both limbs and days. All 95% LOAs were very wide, exceeding the 20N acceptable threshold, ranging from 126.98N (Fig 1, plot C) to 289.09N (Fig 2, plot D).

## Discussion

This is the first study to evaluate test-retest reliability and agreement between a belt-stabilised, handheld MicroFET2 dynamometer and the Forceframe Max system for measuring peak hip flexion and extension force in cyclists, offering novel data. Although both devices initially appear reliable, the observed MDC and agreement values carry potentially important practical and clinical implications.

### Reliability and measurement error

For hip flexion, both devices demonstrated good to excellent within and between-session reliability, and measurement error. Specifically, B-HHD ICCs ranged between 0.77–0.93, with SEMs of 14.25N–22.71N (7.19%−10.38%). These values

**Table 3. Between-session descriptive statistics, ICCs, 95%CIs, SEM and MDC estimates for both devices, according to limb.**

| Between- session Reliability | | Left limb | | | | | | Right limb | | | | | |
|---|---|---|---|---|---|---|---|---|---|---|---|---|---|
| *Measure* | *System* | *Mean* | *SD* | *ICC* | *95% CI* | *SEM (%)* | *MDC (%)* | *Mean* | *SD* | *ICC* | *95% CI* | *SEM (%)* | *MDC (%)* |
| Hip Flexion | B-HHD | 205.96 | 40.80 | 0.81 | 0.53–0.92 | 17.66 (8.57) | 48.95 (23.77) | 222.35 | 55.08 | 0.93 | 0.84 – 0.97 | 14.87 (6.69) | 41.21 (18.53) |
| | FF | 212.40 | 36.68 | 0.84 | 0.61–0.93 | 14.88 (7.01) | 41.25 (19.42) | 222.37 | 40.00 | 0.77 | 0.49–0.90 | 18.98 (8.54) | 52.61 (23.66) |
| Hip Extension | B-HHD | 209.93 | 68.85 | 0.92 | 0.82–0.97 | 19.48 (9.28) | 54.01 (25.73) | 216.07 | 68.41 | 0.90 | 0.78–0.96 | 21.53 (9.96) | 59.67 (27.62) |
| | FF | 246.64 | 74.39 | 0.85 | 0.66–0.93 | 29.05 (11.78) | 80.51 (32.64) | 245.71 | 71.97 | 0.93 | 0.84 – 0.97 | 19.21 (7.82) | 53.26 (21.68) |

Note: SEM and MDC measures are presented in newtons (N), whilst (%) values present the estimates as a proportion relative to mean values. Key: B-HHD = belt stabilised handheld dynamometer; CI = confidence interval; FF = ForceFrame; ICC – intraclass correlation coefficient; MDC = minimal detet-cable change; SEM = standard error of measurement.

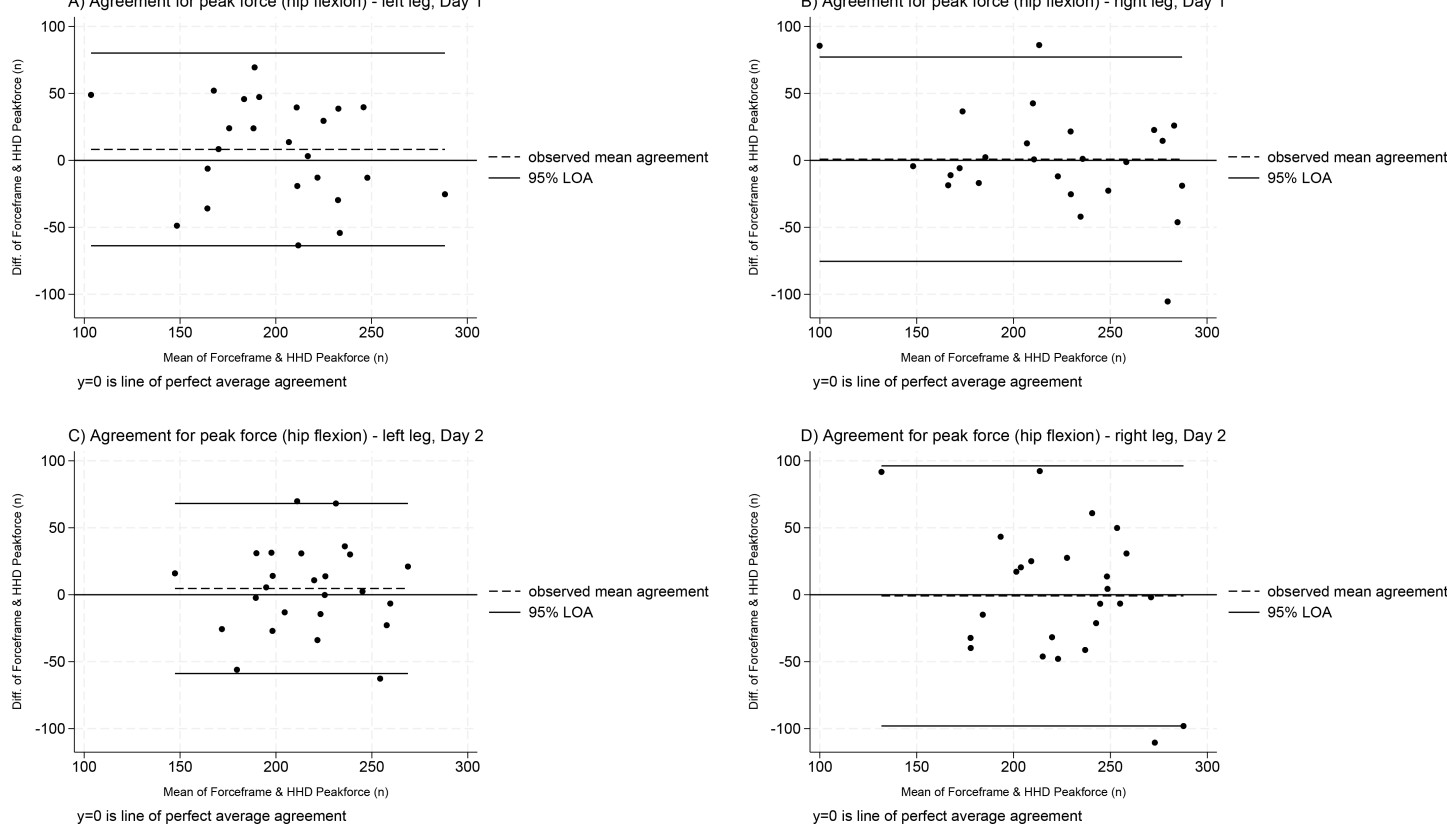

**Fig 1. Within-session agreement between devices for hip flexion peak force measurement (both limbs and testing days).** Note: The Y axis (difference between devices) at 0 indicates perfect agreement; positive values show the ForceFrame recorded greater values, whereas negative values show the Belt-stabilised handheld dynamometer recorded greater values. Thick black lines correspond to 95% limits of agreement, dashed black lines represent observed mean agreement. Key: B-HHD = Belt-stabilised handheld dynamometer; diff = difference, FF = ForceFrame; LOA = limits of agreement; n = newtons.

are generally comparable to a recent meta-analysis of portable, externally stabilised dynamometers for seated hip flexion MVIC assessments, which reported intra-rater WCCs of 0.91 (95%CI = 0.84–0.99) and average SEMs of 15.00N. This similarity was expected, as most devices included in the meta-analysis used comparable B-HHDs, seated testing positions and protocols to those used in our study.

We also found that between-session B-HHD estimates ($ICC_{3.k}$ = 0.81–0.93) were comparable to those recently reported for peak torque by McNabb et al ($ICC_{2.1}$ = 0.91) [30]. However, our B-HHD SEMs (6.69%−8.57%) indicated greater precision than the 14% reported by McNabb et al [30]. While both studies used similar protocols and equipment, this subtle difference may be explained by the distinct constructs measured between studies. While related, peak force (used in our study) is the maximal linear force produced during a task, whereas peak torque (used by McNabb et al [30]) is the maximal rotational force produced during a task [44]. Population differences may have also contributed, as we studied recreational cyclists, rather than a general cohort of healthy adults.

As this is the first study to evaluate the Forceframe Max for hip flexion, direct comparisons to other studies are not possible. A previous meta-analysis reported that externally stabilised dynamometers in supine (as tested in our study) had excellent intra-tester reliability for (WCC = 0.93, 95%CI = 0.89–0.98; average SEMs = 35.50N) [27]. We found that within session FF reliability was consistent with these findings ($ICC_{3.1}$ = 0.89–0.95), but between-session reliability was slightly

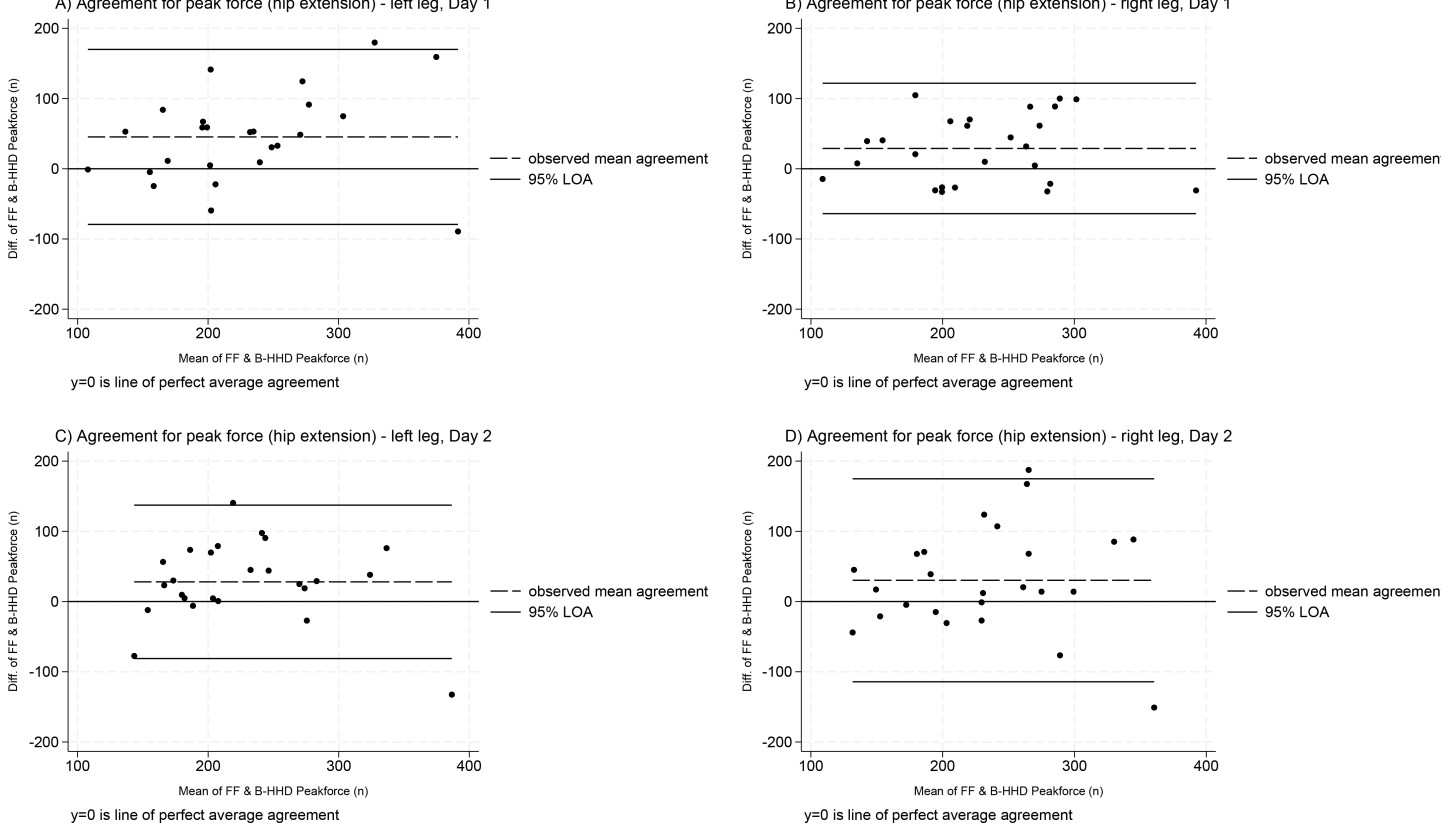

**Fig 2. Within-session agreement between devices for hip extension peak force measurement (both limbs and testing days).** Note: The Y axis (difference between devices) at 0 indicates perfect agreement; positive values show the Forceframe recorded greater values, whereas negative values show the Belt-stabilised handheld dynamometer recorded greater values. Thick black lines correspond to 95% limits of agreement, dashed black lines represent observed mean agreement. Key: B-HHD = Belt-stabilised handheld dynamometer; diff = difference, FF = ForceFrame; LOA = limits of agreement; n = newtons.

lower (ICCs$_{3.k}$ = 0.77–0.84). However, the FF offered greater precision both within and between-sessions (SEM = 7.80 – 18.98N (3.47–8.54%)). This may reflect differences in the devices included in the meta-analysis, which consisted of various B-HHDs and other dynamometers that may not be directly comparable with the FF.

Our results closely aligned with those reported for hip adduction (test-retest ICC$_{2.1}$ = 0.81–0.90; SEM = 6.4–8.6%)) and adduction (test-retest ICC$_{2.1}$ = 0.72–0.92 SEM = 4.3–11.9%) peak force measurment, using a different iteration of the FF in field sport athletes [33]. These findings tentatively suggest that different FF models may perform consistently across populations and hip muscle groups, though further studies are needed to confirm this.

For hip extension, both devices demonstrated good to excellent reliability and similar measurement error (B-HHD ICCs = 0.90–0.95, SEMs = 15.77−21.53N (or 7.38–9.96%); FF ICCs = 0.85–0.95, SEMs = 19.21−29.05N (7.82–11.78%) within and between sessions). These were comparable with values previously reported in a previous meta-analysis of externally stabilised dynamometers used in prone positions (WCC = 0.90; 95%CI = 0.66–0.99; average SEM = 16.90N) [27]. Similarly to hip flexion, we also found B-HHD reliability was consistent with values reported by McNabb et al. [30] for peak torque (ICC$_{2.1}$ = 0.95; SEM = 12.00%), although SEM were marginally less in our study [30].

As before, the FF cannot be compared with previous hip extension data, though our observed values were similar to those previously discussed, from a different ForceFrame iteration for hip abduction and abduction peak force [33].

                                                    

Collectively, these findings suggest that while both devices in our study are reliable for hip extension MVIC peak force measurement, the close consistency with other dynamometers, muscle groups and measurement parameters may be a function of the stable prone test position for hip extension, but this should be also be confirmed in future studies.

## Minimal detectable change and clinical implications

Our observed MDC values for both devices suggest potential practical limitations for both devices. For hip flexion, the FF demonstrated consistently lower within-session MDCs (FF = 21.61N–39.48N (9.6–18.00%); B-HHD = 39.50-62.98N (20.0–28.8%)). However, between-session MDCs were similar (MDC range = 41.21–52.61N) and generally consistent with those reported for all externally stabilised portable dynamometers (MDC = 46.7N–49.3N) [27], but slightly less than 32% observed for peak torque measurement by McNabb et al [30] using a B-HHD. For hip extension, the MDCs observed for both devices (MDCs = 43.71–67.12N (20.50–27.80%) were also broadly comparable with these previous reports [27,30].

MDC values are crucial for distinguishing true performance changes from measurement error [41]. Considering that strength changes of less than 15% are clinically and practically important [26], all within-session B-HHD MDCs, and three quarters of within session FF MDCs were greater than 15% of mean values. Additionally, all between-session MDCs were greater than 15% of mean values for both devices (range = 18.50% to 32.60%). This raises some concerns about the sensitivity of both devices to detect subtle but meaningful changes in performance both within a session (e.g., following physiotherapy interventions such as manual therapy or proprioceptive taping) or over time as a response to rehabilitation or training.

For illustration, isometric training may result in strength improvements of 4.34% per week [45], and 21.5% after 6–12 weeks [46] in healthy adults. Dynamic resistance training can result in strength improvements of 1.77% [47] to 5.20% [48] per week, increasing to between 27.00% and 32.00% after 6 weeks [46,47,49].

Because these weekly adaptations are considerably less than all observed MDC values, it is unlikely that either device could detect true peak force changes if measured on a week-to-week basis. Furthermore, the similarity between six-week adaptations and between-session MDC estimates suggests that that any longer term (i.e., > 6 week) changes should also be cautiously interpreted as these may still be influenced by measurement error.

## Between system agreement

Despite established concurrent validity between the MicroFET2 and Kangatech KT360 [43] (a similar device to the FF), the poor agreement between for hip flexion and extension was surprising, particularly given how far the 95% LOA were exceeded. Possible contributing factors include differences in device resolution (FF = 1.00N; MicroFET B-HHD = 0.44N) [50,51] and the potentially greater stability offered by the FF's rigid aluminium frame and base, compared to the B-HHD's fabric belt. Specifically for hip flexion, differences in testing positions may have introduced additional variation. Theoretically, the seated position used with the B-HHD may be less stable than the supine position used with the FF. However, this did not appear to systematically bias the measurements, as the mean agreement was low (range = −0.88–8.22 N). In contrast, there was consistent, larger bias for hip extension, with the FF recording higher peak force (28.04 to 45.38 N). Importantly, across all measurement scenarios presented in Figs 1 and 2, even adjusting values by the largest recorded mean difference (45.38N), the LOA would still exceed the 20N acceptability threshold, making this an unhelpful transformation.

Due to the observed poor agreement, hip flexion and extension peak force data cannot be interchanged between devices in this population. In practice, this means that measurements should be considered as device-specific; we cautiously recommend against comparing FF data with B-HHD data (and vice versa), and suggest the same device should be used for repeated measurements.

 

## Limitations

Our study has some limitations. The marginal increase in peak force between days suggests a minor learning effect despite familiarisation. While no evidence of fatigue was observed, the validated Rating of Fatigue Scale [52], or other subjective wellbeing scores [53] could have quantified fatigue effects. The second test session was conducted approximately 72 hours after the first, but there was some variability around this due to participant's external commitments, so could be more robustly controlled for in future studies.

Finally, since we used two-way mixed effects (ICC 3.1 and 3.k) models, our findings are not generalisable [38]. Future studies should replicate and confirm our data across different populations. This includes elite cylists, who are likely to utilise regular strength assessments.

## Conclusion

This study found that that the MicroFET2 and Forceframe Max have good to excellent reliability and comparable precision for assessing hip flexor and extensor peak force in recreational cyclists. However, their MDC values mean that these devices have potentially limited clinical and practical usefulness in this population. The poor observed agreement means that interchangeability between devices cannot currently be recommended. Given the two-way mixed effects ICC models used, further studies are recommended in various populations to confirm these findings.

## Supporting information

**S1 Appendix. Sample size calculation.**
(DOCX)

**S2 Appendix. Testing positions.**
(DOCX)

**S3 Appendix. Distribution of raw scores and residuals.**
(DOCX)

**S4 Appendix. Distribution of between device differences.**
(DOCX)

**S5 Appendix. Limits of agreement estimates.**
(DOCX)

## Acknowledgments

The authors would like to thank all staff within the Manchester Metropolitan University Sport team, British Cycling Clubs and Community Groups for their help and support with participant recruitment (without whom this study would not be possible). The authors also thank: Katie Flatters and Heather Stephens at British Cycling for support with study conception; Prof. Michael Callaghan at Manchester Metropolitan University for methodology advice; Dr. Jamie Sergeant at University of Manchester for statistical advice and Liam Parkinson from VALD Performance for providing the VALD ForceFrame Max.

## Author contributions

**Conceptualization:** Dion D'Mello, Benn Digweed, Tom Hughes.

**Data curation:** Dion D'Mello, Tom Hughes.

**Formal analysis:** Dion D'Mello, Tom Hughes.

**Investigation:** Dion D'Mello.

**Methodology:** Dion D'Mello, Benn Digweed, Tom Hughes.

**Project administration:** Dion D'Mello.

**Resources:** Dion D'Mello.

**Software:** Dion D'Mello, Tom Hughes.

**Supervision:** Tom Hughes.

**Visualization:** Dion D'Mello, Benn Digweed, Tom Hughes.

**Writing – original draft:** Dion D'Mello.

**Writing – review & editing:** Dion D'Mello, Benn Digweed, Tom Hughes.

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
