## [Decision Letter · Decision Letter 0]

11 Sep 2025

PONE-D-25-34937The test-retest reliability and agreement between a fixed frame and belt-stabilised handheld dynamometer for isometric hip flexion and extension peak force measurement in recreational cyclistsPLOS ONE

Dear Dr.  Hughes,

Thank you for submitting your manuscript to PLOS ONE. After careful consideration, we feel that it has merit but does not fully meet PLOS ONE’s publication criteria as it currently stands. Therefore, we invite you to submit a revised version of the manuscript that addresses the points raised during the review process.

We look forward to receiving your revised manuscript.

Kind regards,

Sohel Ahmed, BPT, MPT, MDMR

Academic Editor

PLOS ONE

Journal Requirements:

“DDM was a student at Manchester Metropolitan University and this work was submitted as a dissertation project for the qualification of MSc Physiotherapy. He was employed at Manchester Movement Unit and British Cycling as a Sports Massage therapist during data collection. The project was conceived during his employment with British Cycling. BD was employed by Manchester Metropolitan University and the UK Sports Institute. TH is employed at Manchester Metropolitan University.”

Reviewer's Responses to Questions

**Comments to the Author**

1. Is the manuscript technically sound, and do the data support the conclusions?

Reviewer #1: No

Reviewer #2: Yes

2. Has the statistical analysis been performed appropriately and rigorously? 

Reviewer #1: Yes

Reviewer #2: Yes

3. Have the authors made all data underlying the findings in their manuscript fully available?

The PLOS Data policy requires authors to make all data underlying the findings described in their manuscript fully available without restriction, with rare exception (please refer to the Data Availability Statement in the manuscript PDF file). The data should be provided as part of the manuscript or its supporting information, or deposited to a public repository. For example, in addition to summary statistics, the data points behind means, medians and variance measures should be available. If there are restrictions on publicly sharing data—e.g. participant privacy or use of data from a third party—those must be specified.requires authors to make all data underlying the findings described in their manuscript fully available without restriction, with rare exception (please refer to the Data Availability Statement in the manuscript PDF file). The data should be provided as part of the manuscript or its supporting information, or deposited to a public repository. For example, in addition to summary statistics, the data points behind means, medians and variance measures should be available. If there are restrictions on publicly sharing data—e.g. participant privacy or use of data from a third party—those must be specified.requires authors to make all data underlying the findings described in their manuscript fully available without restriction, with rare exception (please refer to the Data Availability Statement in the manuscript PDF file). The data should be provided as part of the manuscript or its supporting information, or deposited to a public repository. For example, in addition to summary statistics, the data points behind means, medians and variance measures should be available. If there are restrictions on publicly sharing data—e.g. participant privacy or use of data from a third party—those must be specified.requires authors to make all data underlying the findings described in their manuscript fully available without restriction, with rare exception (please refer to the Data Availability Statement in the manuscript PDF file). The data should be provided as part of the manuscript or its supporting information, or deposited to a public repository. For example, in addition to summary statistics, the data points behind means, medians and variance measures should be available. If there are restrictions on publicly sharing data—e.g. participant privacy or use of data from a third party—those must be specified.

Reviewer #1: Yes

Reviewer #2: Yes

4. Is the manuscript presented in an intelligible fashion and written in standard English?

Reviewer #1: Yes

Reviewer #2: Yes

5. Review Comments to the Author

Reviewer #1: Study question is clear, but several issues limit the strength of the contribution, particularly in methodology, clarity of presentation, and robustness of conclusions

The Introduction does not clearly highlight the specific gap in the literature that this study addresses.

What does this study add that was not previously known

No sample size/power calculation is provided.

which covariates were adjusted for, rationale for chosen tests, handling of missing data

Tables/figures could be improved for clarity: labels are sometimes unclear; some redundancy exists between text and figures

Conclusions sometimes overstate findings relative to the data presented

Some sentences are overly long and could be simplified.

Reviewer #2: 1) Abstract could better highlight the clinical implications of poor agreement (e.g., why interchangeability matters in practice).

2) Potential bias: hip flexion positions differed (seated vs supine). This introduces systematic variation that may partly explain poor agreement. Needs stronger acknowledgment

3)Assumption of normality is mentioned but no normality test reported—should be clarified.

6. PLOS authors have the option to publish the peer review history of their article (what does this mean?). If published, this will include your full peer review and any attached files.). If published, this will include your full peer review and any attached files.). If published, this will include your full peer review and any attached files.). If published, this will include your full peer review and any attached files.

...

Reviewer #1: **Yes:**Ahmed Ibrahim Al KharusiAhmed Ibrahim Al KharusiAhmed Ibrahim Al KharusiAhmed Ibrahim Al Kharusi

Reviewer #2: **Yes:**Esedullah AKARASEsedullah AKARASEsedullah AKARASEsedullah AKARAS

---

## [Author Response · Author response to Decision Letter 1]

21 Oct 2025

Tom Hughes

Department of Health Professions

Manchester Metropolitan University

Email: t.hughes@mmu.ac.uk

Dear Dr Ahmed

Re: Revisions required for manuscript entitled: The test-retest reliability and agreement between a fixed frame and belt-stabilised handheld dynamometer for isometric hip flexion and extension peak force measurement in recreational cyclists (PONE-D-25-34937)

We would like to take the opportunity to extend our thanks to you and the reviewers for taking the time to consider our manuscript for publication in PLOS ONE. We are encouraged by the favourable response and have taken time to consider the feedback provided. We have now addressed these issues and feel that the manuscript offers a stronger submission as a result.

Please find our replies (in bold type) to the reviewer’s comments (in italic type), which explain how they have been dealt with in the paper. All amended text in the manuscript has been highlighted in yellow. We have copied and pasted excerpts of the text in the replies below where appropriate, to identify the changes made. Please note that line numbers referenced in the response relate to the Revised Manuscript with Tracked Changes.

Editor Comments

Thank you for highlighting this. We have now amended the title page and adjusted formatting precisely throughout the document to ensure the manuscript conforms to publishing guidelines.

“DDM was a student at Manchester Metropolitan University and this work was submitted as a dissertation project for the qualification of MSc Physiotherapy. He was employed at Manchester Movement Unit and British Cycling as a Sports Massage therapist during data collection. The project was conceived during his employment with British Cycling. BD was employed by Manchester Metropolitan University and the UK Sports Institute. TH is employed at Manchester Metropolitan University.”

We apologise for the omission of this statement. As requested, we have included the statement in the ‘Competing Interests’ section (line 627) which now states:

“DDM was a student at Manchester Metropolitan University and this work was submitted as a dissertation project for the qualification of MSc Physiotherapy. He was employed at Manchester Movement Unit and British Cycling as a Sports Massage therapist during data collection. The was conceived during his employment with British Cycling. BD was employed by Manchester Metropolitan University and the UK Sports Institute. TH is employed at Manchester Metropolitan University. This does not alter our adherence to PLOS ONE policies on sharing data and materials.”

Please include your full ethics statement in the ‘Methods’ section of your manuscript file. In your statement, please include the full name of the IRB or ethics committee who approved or waived your study, as well as whether or not you obtained informed written or verbal consent. If consent was waived for your study, please include this information in your statement as well.

Thank you for your comment and we apologise if this was unclear in the submitted document. This was already in place in the ‘Methods’ section, under the subheading of ‘Participants’. For clarity, we have also added that the study was conducted in accordance with the Declaration of Helskini in 2013, as data collection preceded publication of the updated Declaration on 19 October 2024. This text now states (line 214):

“Ethical approval was granted by the Faculty of Health and Education Research Ethics and Governance Committee at Manchester Metropolitan University (no.59674) and conducted in accordance with the 2013 Declaration of Helsinki. All participants provided written informed consent prior to participation.”

The reviewers have not recommended any citations to be added, so no action is needed here.

Please review your reference list to ensure that it is complete and correct. If you have cited papers that have been retracted, please include the rationale for doing so in the manuscript text or remove these references and replace them with relevant current references. Any changes to the reference list should be mentioned in the rebuttal letter that accompanies your revised manuscript. If you need to cite a retracted article, indicate the article’s retracted status in the References list and also include a citation and full reference for the retraction notice.

All references included have been checked again for completeness on 8th October 2025. No papers included have a retracted status. In relation to addressing a comment from Reviewer 1, we have added in a new reference (see below)

When completing the data availability statement of the submission form, you indicated that you will make your data available on acceptance. We strongly recommend all authors decide on a data sharing plan before acceptance, as the process can be lengthy and hold up publication timelines. Please note that, though access restrictions are acceptable now, your entire data will need to be made freely accessible if your manuscript is accepted for publication. This policy applies to all data except where public deposition would breach compliance with the protocol approved by your research ethics board. If you are unable to adhere to our open data policy, please kindly revise your statement to explain your reasoning and we will seek the editor's input on an exemption. Please be assured that, once you have provided your new statement, the assessment of your exemption will not hold up the peer review process.

Apologies for this; our data sharing plan has already been agreed, hence the reason why we have included our raw data and code; this has now been amended on the submission form.

Reviewer 1 Comments

Study question is clear, but several issues limit the strength of the contribution, particularly in methodology, clarity of presentation, and robustness of conclusions

The Introduction does not clearly highlight the specific gap in the literature that this study addresses.

What does this study add that was not previously known?

Thank you for taking the time and effort to review our paper, and we appreciate your feedback.

In relation to your point made above, throughout the introduction, we have explicitly outlined strong reasons for why this study is needed.

For example, on line 135, we highlight that because hip strength influences cycling performance, strength evaluation is important:

“Given the importance of hip strength in cycling performance and the risk of injury associated with cycling participation (18), regular muscle strength assessments may provide valuable data for cyclists, coaches and health professionals. Indeed, regular physical testing can be useful for performance tracking, training evaluation and planning (19-21), setting rehabilitation targets (19) and assessing treatment effectiveness following injury (22).”

On line 159, to justify investigating the use of belt stabilised handheld dynamometers (B-HHDs) in cyclists we explicitly state that:

“Importantly, population characteristics influence device reliability (31, 32), but to date, the reliability of external, belt-stabilised HHDs (B-HHDs) has not been investigated in cyclists, so it is unclear if these estimates apply to this population.”

We also justify the need to investigate fixed frame ForceFrame (FF) devices in cyclists (line 164):

“While belt stabilisation may improve reliability, newer fixed frame dynamometers (such as the Vald ForceFrame (herein termed FF) Max (Vald Performance, Brisbane, Australia)), may offer superior stabilisation because sensors are mounted directly to a frame and base. Fixed frame devices are also highly adjustable, which allows testing of most muscle groups. A different iteration of the FF (i.e. ForceFrame Fold) has shown good to excellent test-retest reliability for hip adduction (ICC2.1= 0.81-0.90) and moderate to excellent for hip abduction MVICs (ICC 2.1= 0.72-0.92) in field sport athletes (33). However, to the best of our knowledge, no previous studies have investigated the reliability of any FF version for measuring hip flexion and extension MVICs in any population.”

Finally, given the increased use of fixed frame devices to replace traditionally used B-HHDs in practice, we specifically state that (line 175):

“Given the growing preference to replace HHDs with fixed frame devices, (34) it is essential to determine whether measurements from both devices can be interchanged. This may help to ensure consistent and accurate data in settings where both devices are available, where longitudinal data have already been collected by B-HHDs, or when practitioners are deciding whether to use one method over another.

The aim of this study was to evaluate the test-retest reliability and agreement between a belt-stabilised MicroFET2 HHD and the ForceFrame Max, for measurement of hip flexion and extension MVIC peak force in recreational cyclists.”

Consequently, we feel that we have strongly justified and reasoned why this study is needed, and why its novelty will contribute to the existing evidence base. We therefore feel no change is necessary here.

No sample size/power calculation is provided.

Thank you for your comment. However, we have already specifically provided a sample size calculation (S1 Appendix), and there is a dedicated explanation of sample size in the Methods section on line 219, which states:

“Sample size was calculated using the confidence interval (CI) width procedure of the ICC & SEM Power Sample Size Decision Assistant (https://iriseekhout.shinyapps.io/ICCpower) (38). Previously reported intraclass correlation coefficents (ICCs) for hip flexion and extension strength measurements were 0.91 and 0.90 for HHDs with external stabilisation (27) and 0.77- 0.86 for the Groinbar, (Vald Performance, Brisbane, Australia) (23) a similar fixed frame device to the Forceframe Max used in this study. For a two-way mixed effects ICC model needed for a test-retest design, our calculation assumed a conservative expected correlation of 0.80, a moderate variance of 10 for between-participant test scores with no expected systematic differences, and aimed for a target CI width of 0.30. Our calculation indicated that recruitment of 25 participants would enable ICC estimation with CI precision of 0.26 for within-session analyses (three trials per testing session), and 0.33 for between-session analyses (mean of three trials are compared over two testing sessions) (see S1 Appendix for full calculation).”

Therefore, we feel no further action is required.

Which covariates were adjusted for, rationale for chosen tests, handling of missing data

Thank you for your comment. In accordance with the COnsensus-based Standards for the selection of health Measurement INstruments (COSMIN) Reporting Guideline (Gagnier et al, 2021), COSMIN Risk of Bias Tool (Mokkink et al, 2021) and its Guidance for Systematic Reviews of Patient-Reported Outcome Measures (Mokkink et al, 2024), adjustment for covariates is neither required nor recommended in studies assessing reliability. COSMIN advocates for the use of intraclass correlation coefficients (ICCs) to evaluate reliability for continuous data, which we deemed appropriate for our analysis. ICCs are designed to capture the proportion of total variance attributable to differences between subjects or clusters. Introducing covariate adjustment can artificially reduce this variance, resulting in context-specific estimates that may not accurately reflect the underlying reliability of the measure. For this reason, we followed COSMIN’s recommendation to report unadjusted ICCs. Adjusted ICCs may be appropriate in exploratory analyses or when assessing sources of heterogeneity, but they are not standard for primary reliability reporting.

In terms of the rationale for ICC model selection, we apologise that this is unclear. We have amended this and added a reference to support our choice (Koo and Li, 2016). This can be found in two locations: line 226, which now states:

“For a two-way mixed effects ICC model that is most suitable for a test-retest design used in this study , our calculation assumed a conservative expected correlation of 0.80, a moderate variance of 10 for between-participant test scores with no expected systematic differences, and aimed for a target CI width of 0.30.”

This was also reiterated this in the statistical analysis section (line 304):

“Two-way mixed effects ICC models are required for a test-retest design (39) used in this study, so within session ICCs3.1 and between session ICCs3.k were calculated with corresponding 95% CIs and interpreted as poor < 0.40; fair= 0.40- 0.70; good=0.70-0.90; excellent > 0.90 (40)”.

In terms of missing data, only one particpant withdrew and this is clearly stated in the results, but we have specifically amended this to clarify that there is no other missing data (line 326):

“Of twenty-six participants recruited, one withdrew due to pain during testing and was excluded from the analysis. In the remaining dataset, there were no instances of missing data.”

Tables/figures could be improved for clarity: labels are sometimes unclear; some redundancy exists between text and figures

Thank you for highlighting this. We agree that the units of measurement could be clearer. We have now amended the captions for Table 1, which now states (line 346):

“Table 1: Within-session descriptive statistics for peak force measurements, per device and testing day.”

The caption now states (line 352):

“Key: B-HHD = belt stabilised handheld dynamometer; FF = Forceframe; SD = standard deviation. Please note that all measurements are in newtons (N).”

The title of Table 2 has also been clarified and now states (Line 361):

“Table 2: Within-session ICCs3.1, 95%CIs, SEM and MDC estimates for both devices and limbs, according to testing day”.

Table 3 has also been clarified; on review of the submitted manuscript there appears to have been an issue with the text alignment. This has now been resolved and the title has now been amended, which now states (line 396):

“Table 3: Between-session descriptive statistics, ICCs, 95%CIs, SEM and MDC estimates for both devices, according to limb.”

Conclusions sometimes overstate findings relative to the data presented

Thank you for your feedback. We have taken care to try not to overstate the findings of the study, and this is especially important given the restricted generalisability due to the two-way mixed ICC models used.

Examples of where we have been careful with terminology include:

Line 496: “These findings tentatively suggest that different FF models may perform consistently across populations and hip muscle groups, though further studies are needed to confirm this.”

Line 514: “Collectively, these findings suggest that while both devices in our study are reliable for hip extension MVIC peak force measurement, the close consistency with other dynamometers, muscle groups and measurement parameters may be a function of the stable prone test position for hip extension, but

---

## [Decision Letter · Decision Letter 1]

12 Nov 2025

The test-retest reliability and agreement between a fixed frame and belt-stabilised handheld dynamometer for isometric hip flexion and extension peak force measurement in recreational cyclists

PONE-D-25-34937R1

Dear Dr. Hughes,

We’re pleased to inform you that your manuscript has been judged scientifically suitable for publication and will be formally accepted for publication once it meets all outstanding technical requirements.

Kind regards,

Sohel Ahmed, BPT, MPT, MDMR

Academic Editor

PLOS ONE

Additional Editor Comments (optional):

Reviewers' comments:

Reviewer's Responses to Questions

**Comments to the Author**

1. If the authors have adequately addressed your comments raised in a previous round of review and you feel that this manuscript is now acceptable for publication, you may indicate that here to bypass the “Comments to the Author” section, enter your conflict of interest statement in the “Confidential to Editor” section, and submit your "Accept" recommendation.

Reviewer #1: All comments have been addressed

Reviewer #2: All comments have been addressed

2. Is the manuscript technically sound, and do the data support the conclusions?

Reviewer #1: Yes

Reviewer #2: Yes

3. Has the statistical analysis been performed appropriately and rigorously? 

Reviewer #1: Yes

Reviewer #2: Yes

4. Have the authors made all data underlying the findings in their manuscript fully available?

The PLOS Data policy requires authors to make all data underlying the findings described in their manuscript fully available without restriction, with rare exception (please refer to the Data Availability Statement in the manuscript PDF file). The data should be provided as part of the manuscript or its supporting information, or deposited to a public repository. For example, in addition to summary statistics, the data points behind means, medians and variance measures should be available. If there are restrictions on publicly sharing data—e.g. participant privacy or use of data from a third party—those must be specified.requires authors to make all data underlying the findings described in their manuscript fully available without restriction, with rare exception (please refer to the Data Availability Statement in the manuscript PDF file). The data should be provided as part of the manuscript or its supporting information, or deposited to a public repository. For example, in addition to summary statistics, the data points behind means, medians and variance measures should be available. If there are restrictions on publicly sharing data—e.g. participant privacy or use of data from a third party—those must be specified.requires authors to make all data underlying the findings described in their manuscript fully available without restriction, with rare exception (please refer to the Data Availability Statement in the manuscript PDF file). The data should be provided as part of the manuscript or its supporting information, or deposited to a public repository. For example, in addition to summary statistics, the data points behind means, medians and variance measures should be available. If there are restrictions on publicly sharing data—e.g. participant privacy or use of data from a third party—those must be specified.requires authors to make all data underlying the findings described in their manuscript fully available without restriction, with rare exception (please refer to the Data Availability Statement in the manuscript PDF file). The data should be provided as part of the manuscript or its supporting information, or deposited to a public repository. For example, in addition to summary statistics, the data points behind means, medians and variance measures should be available. If there are restrictions on publicly sharing data—e.g. participant privacy or use of data from a third party—those must be specified.

Reviewer #1: Yes

Reviewer #2: Yes

5. Is the manuscript presented in an intelligible fashion and written in standard English?

Reviewer #1: Yes

Reviewer #2: Yes

6. Review Comments to the Author

Reviewer #1: Thank you for resending the review. All Comments have been addressed and nothing to add to the previous comments.

Reviewer #2: (No Response)

7. PLOS authors have the option to publish the peer review history of their article (what does this mean?). If published, this will include your full peer review and any attached files.). If published, this will include your full peer review and any attached files.). If published, this will include your full peer review and any attached files.). If published, this will include your full peer review and any attached files.

...

Reviewer #1: **Yes:**Ahmed Ibrahim Al KharusiAhmed Ibrahim Al KharusiAhmed Ibrahim Al KharusiAhmed Ibrahim Al Kharusi

Reviewer #2: **Yes:**Esedullah AKARASEsedullah AKARASEsedullah AKARASEsedullah AKARAS

---

## [Editor Report · Acceptance letter]

PONE-D-25-34937R1

PLOS One

Dear Dr. Hughes,

I'm pleased to inform you that your manuscript has been deemed suitable for publication in PLOS One. Congratulations! Your manuscript is now being handed over to our production team.

Kind regards,

on behalf of

Dr. Sohel Ahmed

Academic Editor

PLOS One